# Clinicopathological Features and Risk Stratification of Multiple-Classifier Endometrial Cancers: A Multicenter Study from Poland

**DOI:** 10.3390/cancers17152483

**Published:** 2025-07-28

**Authors:** Wiktor Szatkowski, Małgorzata Nowak-Jastrząb, Tomasz Kluz, Aleksandra Kmieć, Małgorzata Cieślak-Steć, Magdalena Śliwińska, Izabela Winkler, Jacek Tomaszewski, Jerzy Jakubowicz, Renata Pacholczak-Madej, Paweł Blecharz

**Affiliations:** 1Department of Gynaecologic Oncology Maria Sklodowska-Curie National Research Institute of Oncology Krakow Branch, 31-315 Krakow, Poland; malgorzata.nowak@krakow.nio.gov.pl (M.N.-J.); jerzy@jakubowicz.pl (J.J.); renata.pacholczak@gmail.com (R.P.-M.); pawel.blecharz@interia.pl (P.B.); 2Department of Gynecology and Obstetrics, Institute of Medical Sciences, Medical College of Rzeszow University, 35-055 Rzeszów, Poland; jtkluz@interia.pl (T.K.);; 3Department of III Clinic of Radiotherapy and Chemotherapy Maria Sklodowska-Curie National Research Institute of Oncology Gliwice Branch, 44-102 Gliwice, Poland; 4Second Department of Gynecological Oncology, St. John’s Center of Oncology of the Lublin Region, 20-090 Lublin, Poland; ikochans@interia.pl (I.W.); jtomaszewski@cozl.pl (J.T.)

**Keywords:** endometrial carcinoma, molecular classification, multiple-classifier, POLEmut, MMRd, p53abn, triple classifier, ProMisE, ESGO

## Abstract

Endometrial cancers with more than one molecular alteration—so-called multiple-classifier tumors, such as MMRd with p53 abnormalities or POLEmut with p53abn—are identified in 3–11% of patients. According to current classification, these tumors are assigned to a single dominant molecular group, but emerging evidence suggests that coexisting p53 abnormalities may influence their clinical behavior. In this multicenter study involving over 1000 patients, we found that multiple-classifier tumors—especially MMRd-p53abn and POLEmut-p53abn—were more frequently classified as high–intermediate or high-risk according to ESGO/ESTRO/ESP criteria. Moreover, lymph node metastases were more common in POLEmut-p53abn and POLEmut-MMRd-p53abn cases. These findings highlight the need for careful interpretation of tumors with overlapping molecular features and support further research to refine risk stratification and optimize treatment strategies in this unique subgroup of endometrial cancer patients.

## 1. Introduction

Endometrial cancer (EC) is the most common gynecologic malignancy in developed countries, with over 417,000 new cases and nearly 100,000 deaths reported worldwide in 2020 [1,2]. Its incidence continues to rise globally, driven by aging populations, increasing prevalence of obesity, metabolic syndrome, and declining fertility rates [3,4,5]. While early-stage EC is often diagnosed at an early stage and generally carries a favorable prognosis, advanced or recurrent disease remains a significant therapeutic challenge, with a five-year survival rate below 20% [6].

Traditionally, risk stratification in EC relied on histopathological features such as FIGO stage, histological subtype, tumor grade, and the presence of lymphovascular space invasion (LVSI). However, these parameters are subject to interobserver variability and have limited predictive accuracy for guiding personalized therapy [7,8].

The molecular classification proposed by The Cancer Genome Atlas (TCGA) in 2013 revolutionized the understanding of EC by identifying four distinct molecular subgroups: POLE ultramutated (POLEmut), mismatch repair deficient (MMRd), p53 abnormal (p53abn), and no specific molecular profile (NSMP) [5,7,9]. These subtypes have distinct prognostic and therapeutic implications and were later adapted for clinical use through the Proactive Molecular Risk Classifier for Endometrial Cancer (ProMisE) algorithm. The incorporation of molecular subtyping into international guidelines, including those of the European Society of Gynaecological Oncology/European Society for Radiotherapy and Oncology/European Society of Pathology (ESGO/ESTRO/ESP, 2021), the European Society for Medical Oncology (ESMO), and the International Federation of Gynecology and Obstetrics (FIGO, 2023), underscores its relevance in routine practice [10,11,12,13].

Despite these advances, approximately 3–11% of ECs demonstrate features of more than one molecular subtype, referred to as multiple-classifier or mixed-profile tumors (e.g., MMRd–p53abn, POLEmut–p53abn, or POLEmut–MMRd–p53abn) [14,15]. These cases present a unique clinical and diagnostic challenge, as they may harbor conflicting prognostic signals, such as the favorable outlook associated with POLEmut versus the poor prognosis linked to p53abn alterations. In particular, the MMRd–p53abn subgroup has generated substantial interest, with conflicting data on its optimal risk classification and true biological behavior [14,15,16].

Furthermore, accurate characterization of these multiple-classifier tumors requires integration of both immunohistochemical and molecular techniques, as discrepancies can arise between p53 immunohistochemistry and TP53 sequencing, especially in cases with subclonal or ambiguous staining patterns [17,18].

Research efforts on molecular profiling in EC have predominantly focused on Western European and North American populations, with limited data from Central and Eastern Europe. Regional differences in genetic backgrounds, diagnostic protocols, and access to advanced molecular testing could impact the applicability and generalizability of current risk models in these populations.

To address this gap, we conducted a multicenter study involving four leading gynecologic oncology centers in Poland to assess the prevalence and detailed clinicopathological characteristics of multiple-classifier ECs, with a particular emphasis on MMRd–p53abn tumors. By elucidating the morphological and molecular profiles of these rare but clinically relevant subgroups, our findings aim to contribute to ongoing efforts to refine risk stratification and guide future development of individualized treatment strategies.

## 2. Materials and Methods

### 2.1. Study Design and Population

This multicenter retrospective study included 1075 patients with newly diagnosed endometrial cancer (EC) who underwent surgical treatment between April 2022 and March 2025 in four Polish gynecologic oncology centers (Silesian, Lesser Poland, Subcarpathian, and Lublin Voivodeships). Histotype and tumor grade were assigned according to the 2020 WHO classification [19]. Lymphovascular space invasion (LVSI) was classified as absent (no vessels), focal (1–4 vessels), or substantial (≥5 vessels). Histopathological assessment was conducted independently at each center according to WHO criteria, without central pathology review.

Lymph node status was evaluated in 794 of 1075 patients (73.9%). Among these, 552 (69.5%) underwent sentinel lymph node mapping, and 242 (30.5%) underwent systematic pelvic lymphadenectomy. In the remaining 281 patients (26.1%), lymph node evaluation was omitted based on low-risk criteria or clinical judgment.

High–intermediate risk (HIR) and high-risk (HR) groups were defined per ESGO/ESTRO/ESP 2021 guidelines, based exclusively on clinicopathological parameters (e.g., FIGO stage, histotype, grade, LVSI, lymph node involvement), independent of molecular classification to avoid bias from multiple-classifier overlap [11]. Central pathology review was not feasible due to logistical constraints, but adherence to WHO 2020 criteria across centers ensured consistency in histopathological assessments.

### 2.2. Molecular Classification and Immunohistochemistry (IHC)

Tumor tissue was subjected to immunohistochemical evaluation of mismatch repair (MMR) proteins (MLH1, MSH2, MSH6, PMS2) and p53 using the Ventana BenchMark Ultra platform (Roche Diagnostics, Indianapolis, IN, USA). Standardized staining protocols were applied across institutions. MMR deficiency (MMRd) was defined as the loss of at least one MMR protein. Abnormal p53 expression (p53abn) included diffuse overexpression (>80% of tumor cells), complete absence, or subclonal staining, according to ProMisE criteria [20].

### 2.3. DNA Extraction and POLE Mutation Analysis

DNA was extracted from formalin-fixed, paraffin-embedded (FFPE) tissue samples using either the QIAamp DSP DNA FFPE Tissue Kit (Qiagen, Hilden, Germany) or the Maxwell^®^ RSC DNA FFPE Kit (Promega, Madison, WI, USA). POLE exon 9, 11, 13, and 14 sequencing was initially performed using Sanger sequencing (BigDye Terminator v3.1, Applied Biosystems), as this method provides a cost-effective and reliable approach to detect pathogenic mutations in the exonuclease domain. In selected cases with discordant or inconclusive findings—defined as ambiguous MMR or p53 immunohistochemistry (IHC) staining (e.g., partial loss, borderline subclonal p53), discrepancies with clinicopathological features, or suspected multiple-classifier profiles requiring further validation—next-generation sequencing (NGS) was performed.

NGS was conducted on the IonTorrent platform using a custom-designed AmpliSeq panel targeting POLE, TP53, MLH1, MSH2, MSH6, and PMS2 genes. This custom panel was specifically developed to support endometrial cancer molecular classification and ensure high sensitivity for clinically relevant variants. Only pathogenic or likely pathogenic variants (as classified by ClinVar, OncoKB, and Varsome databases) were included in molecular classification [21,22,23].

The same molecular testing protocols and variant interpretation standards were previously applied in a national analysis of endometrial cancer molecular classification practices in Poland, supporting consistency and reproducibility across studies [24].

### 2.4. Statistical Analysis

Statistical analyses were performed using SPSS version 27 (IBM Corp., Armonk, NY, USA). Categorical variables were compared using Fisher’s exact test (for expected cell counts < 5) or chi-square test (for ≥5). Continuous variables were assessed using Student’s *t*-test. Logistic regression was used to estimate odds ratios (ORs) with 95% confidence intervals (CIs) for binary outcomes. Statistical significance was set at *p* < 0.05.

### 2.5. Ethical Considerations

This study was approved by the Institutional Review Board of the Maria Sklodowska-Curie National Research Institute of Oncology, Krakow Branch (Approval No. 6/2025), and conducted in accordance with the Declaration of Helsinki. Informed consent was obtained from all participants. Data collection complied with EU General Data Protection Regulation (GDPR) standards.

## 3. Results

### 3.1. Prevalence of Multiple-Classifier Tumors

Among 1075 patients, multiple-classifier endometrial cancers (ECs) accounted for 6.9% (74/1075), including MMRd-p53abn (3.9%, 42/1075), POLEmut-p53abn (0.4%, 4/1075), POLEmut-MMRd (1.7%, 18/1075), and POLEmut-MMRd-p53abn (0.9%, 10/1075).

### 3.2. MMRd, p53abn, and MMRd-p53abn Comparison

Table 1 presents the clinicopathological characteristics of patients with endometrial cancer by molecular subtype. Compared to classical MMRd tumors (N = 246), MMRd-p53abn tumors (N = 42) had higher rates of non-endometrioid histology (11.90% vs. 2.85%, *p* = 0.018; OR = 3.78, 95% CI: 1.19–12.02), more frequent high-grade (G3) tumors (28.57% vs. 11.79%, *p* = 0.002; OR = 3.26, 95% CI: 1.55–6.86), and a greater proportion assigned to high–intermediate/high-risk (HIR/HR) groups (59.52% vs. 37.80%, *p* = 0.001; OR = 2.81, 95% CI: 1.50–5.25). These comparisons are visually represented in Figure 1 (histotype comparison), Figure 2 (grade comparison), and Figure 3 (risk group comparison).

Compared to p53abn tumors (N = 156), MMRd-p53abn tumors had a lower proportion of non-endometrioid histology (11.90% vs. 35.26%, *p* = 0.001). MMRd-p53abn tumors also trended toward more frequent advanced FIGO III–IV stages (23.81% vs. 13.82%, *p* = 0.192), although this did not reach statistical significance.

### 3.3. POLEmut and Multiple-Classifier POLEmut Comparison

Table 2 presents the clinicopathological comparison of classical POLEmut tumors with multiple-classifier subgroups involving POLE mutations. Compared to classical POLEmut tumors (N = 30), the POLEmut–p53abn subgroup (N = 4) demonstrated significantly higher rates of G3 tumors (75.0% vs. 6.7%, *p* = 0.005; OR = 42.00, 95% CI: 2.87–614.8), lymph node metastases (50.0% vs. 3.3%, *p* = 0.013; OR = 29.00, 95% CI: 1.77–475.3), and FIGO stage III–IV disease (75.0% vs. 6.7%, *p* = 0.005; OR = 42.00, 95% CI: 2.87–614.8). However, interpretation is limited by the small sample size and absence of outcome data.

The POLEmut–MMRd subgroup (N = 18) also showed higher, though not statistically significant, rates of G3 tumors (16.7% vs. 6.7%, *p* = 0.198; OR = 2.80, 95% CI: 0.47–16.62), lymph node metastases (5.6% vs. 3.3%, *p* = 1.000; OR = 1.71, 95% CI: 0.11–27.48), and advanced FIGO stages (11.1% vs. 6.7%, *p* = 1.000; OR = 1.75, 95% CI: 0.23–13.26) relative to classical POLEmut.

Lastly, POLEmut–MMRd–p53abn tumors (N = 10) exhibited more adverse features, including higher rates of G3 tumors (20.0% vs. 6.7%, *p* = 0.247; OR = 3.50, 95% CI: 0.46–26.46), lymph node involvement (30.0% vs. 3.3%, *p* = 0.192; OR = 12.44, 95% CI: 1.05–147.88), and FIGO III–IV stage (40.0% vs. 6.7%, *p* = 0.033; OR = 9.33, 95% CI: 1.23–70.66).

Although these trends suggest a deviation from the favorable prognosis associated with classical POLEmut tumors, the small numbers and lack of survival data preclude definitive conclusions. Survival data were not collected due to the short follow-up period (2022–2025).

## 4. Discussion

Endometrial cancer staging and classification have evolved to integrate traditional pathology with molecular findings, improving risk stratification and treatment personalization. In the era of precision medicine, identifying molecular subtypes and biomarkers (such as POLE mutations or MMR deficiency) is crucial for guiding targeted therapies and advancing patient outcomes [16,18]. Approximately 25–30% of endometrial cancers exhibit mismatch repair deficiency (MMRd) or microsatellite instability (MSI) [25]. These molecular insights have been incorporated into current ESGO/ESTRO/ESP and FIGO guidelines, aiming to refine risk stratification and optimize treatment pathways [11,13].

However, there is still no consensus on how to categorize or manage multiple-classifier tumors, such as MMRd–p53abn or POLEmut–p53abn. Some studies have excluded these tumors from survival analyses, while others have grouped them into one of the four main molecular classes without addressing their unique clinicopathological or biological characteristics [12].

This is the first multicenter study from Central–Eastern Europe analyzing multiple-classifier endometrial cancers (ECs), with a 6.9% prevalence, consistent with the 3–11% range reported globally [14,15,26]. Our higher rate compared to De Vitis et al. (4.8%) [15] may reflect the large cohort size (N = 1075), regional genetic variations, or differences in molecular testing protocols, underscoring the need for standardized diagnostics.

León-Castillo et al. [14] reported favorable recurrence-free survival for MMRd-p53abn endometrial cancers (ECs), similar to MMRd-only tumors (92.2% vs. 70.8% for p53abn; *p* = 0.024). In contrast, our study (N = 1075) found that MMRd-p53abn tumors (3.9%, N = 42) exhibit a distinct morphological profile with more adverse clinicopathological features compared to MMRd-only tumors (N = 246), including the following:Non-endometrioid histology: 11.9% vs. 2.85% (OR = 3.78, 95% CI: 1.19–12.02, *p* = 0.018);Grade 3 tumors: 28.6% vs. 11.8% (OR = 3.26, 95% CI: 1.55–6.86, *p* = 0.002);High–intermediate/high-risk (HIR/HR) status per ESGO/ESTRO/ESP 2021 guidelines: 59.5% vs. 37.8% (OR = 2.81, 95% CI: 1.50–5.25, *p* = 0.001).

These findings suggest that MMRd-p53abn tumors may have a more aggressive clinicopathological profile than MMRd-only tumors, potentially warranting further prognostic evaluation to determine their clinical behavior.

Compared to p53abn tumors, MMRd–p53abn ECs showed fewer non-endometrioid types (11.9% vs. 35.3%; *p* = 0.001), suggesting a distinct molecular profile. Bogani et al. [27] similarly reported an increased recurrence risk in MMRd–p53abn tumors, further supporting the need for reevaluation of this subgroup.

Similarly, both POLEmut–p53abn (N = 4) and POLEmut–MMRd–p53abn (N = 10) tumors demonstrated aggressive features, particularly regarding advanced-stage tumors:FIGO III–IV in 75% (POLEmut-p53abn) and 40% (POLEmut-MMRd-p53abn), compared to 6.7% in classical POLEmut ECs [28];Grade 3 tumors in 75% vs. 6.7% (OR = 42.00; *p* = 0.005);Lymph node metastases in 50% vs. 3.3% (OR = 29.00; *p* = 0.013).

These data raise concern that coexisting p53 abnormalities may negate the typically favorable prognosis of POLEmut tumors. Our findings are in line with observations from Jamieson et al. [29], who reported 14% nodal involvement in POLEmut ECs overall, highlighting that even POLEmut tumors can exhibit aggressive behavior—possibly intensified by the presence of additional alterations such as TP53 mutations.

Furthermore, our results suggest that the aggressive features observed in MMRd–p53abn tumors may have implications for adjuvant treatment planning and immunotherapy eligibility. Due to their MMRd component, these tumors may theoretically benefit from immune checkpoint inhibitors, such as dostarlimab, as demonstrated in the GARNET trial by Mirza et al. [30]. However, the coexisting p53abn alteration may influence tumor behavior and potentially affect treatment response—although the direction and magnitude of this effect remain unclear. This uncertainty underscores the need for prospective trials specifically designed to evaluate immunotherapy efficacy in multiple-classifier ECs.

Several lines of evidence support the hypothesis that p53 abnormalities may exert a dominant negative prognostic effect even within MMRd tumors. Kato et al. [28] showed that in non-Lynch MMRd ECs, p53abn was associated with significantly worse 5-year overall survival (53.6% vs. 93.9%; *p* = 0.0016). Similarly, Michalova et al. [17] reported that among 12 multiple-classifier ECs, most of the cases with available follow-up (6/9) behaved aggressively. In the POLEmut subgroup, 3/4 tumors were advanced stage, with one patient dying of disease. Among MMRd/TP53mut cases, 3/5 developed metastatic disease, and one patient died. De Vitis et al. [15] also observed a trend toward increased recurrence in MMRd–p53abn tumors, although this did not reach statistical significance.

An ongoing debate concerns the role of next-generation sequencing (NGS) in molecular classification of endometrial cancer, as it is not yet a routine component of the ProMisE algorithm. Although immunohistochemistry (IHC) remains the standard method for assessing both p53 status and mismatch repair (MMR) protein expression, discrepancies between IHC and NGS have been increasingly recognized in the literature. These discrepancies may also reflect tumor heterogeneity or mosaic mutations, where subclonal populations harbor distinct molecular alterations not uniformly detected by targeted NGS panels [31,32]. These inconsistencies may stem from differences in detection thresholds, interpretation criteria, or underlying biological mechanisms. In our cohort, such discordances were identified in 11.9% of MMRd–p53abn tumors and may be attributed to splice-site mutations, large insertions/deletions (indels), or subclonal immunostaining—events that can be missed by targeted NGS panels. These technical limitations underscore the importance of integrating both IHC and NGS approaches to enhance the accuracy of molecular classification and to better understand the biological behavior of tumors with multiple concurrent molecular alterations.

Taken together, our findings highlight that multiple-classifier ECs—especially those involving p53 abnormalities—may not align well with existing risk stratification systems. Clinical decisions regarding adjuvant therapy should be guided cautiously, with thorough consideration of stage, grade, and molecular complexity [25]. Future multicenter studies with larger patient cohorts and long-term survival data are essential to refine classification frameworks and to optimize individualized treatment strategies.

## 5. Limitations

This study has several limitations. First, the absence of overall survival (OS) and progression-free survival (PFS) data precludes direct prognostic evaluation of multiple-classifier endometrial cancers (ECs). Second, the relatively small size of some molecular subgroups—particularly POLEmut–p53abn and POLEmut–MMRd–p53abn—limits statistical power and the generalizability of their clinicopathological features. Third, the lack of central pathology review may have introduced interobserver variability in histotype, tumor grade, and LVSI assessment, although standardized WHO 2020 criteria were applied at all participating centers. Fourth, detailed characterization of MMR protein loss (e.g., isolated vs. paired MLH1/PMS2 or MSH2/MSH6 loss) was not performed, which may limit the interpretability of MMRd subtypes. Molecular testing was conducted using harmonized protocols, but variation in the use of NGS (only in selected cases) may have affected the detection of rare multiple-classifier combinations.

In addition, the quality of formalin-fixed paraffin-embedded (FFPE) tissue samples, as well as potential DNA degradation, may have impacted the sensitivity of both NGS and IHC, particularly for detecting low-frequency or subclonal variants. Furthermore, inter-institutional variability in IHC interpretation or molecular testing protocols, despite harmonization efforts, may have contributed to minor inconsistencies in molecular classification. Finally, multivariate regression was not performed due to the small size of some molecular subgroups and incomplete data on comorbidities, which limited the feasibility of adjusting for confounding factors such as age and FIGO stage.

## 6. Conclusions

Multiple-classifier ECs, particularly MMRd–p53abn, POLEmut–p53abn, and POLEmut–MMRd–p53abn, appear to exhibit distinct clinicopathological features compared to single-classifier tumors. The presence of p53 abnormalities—even in tumors harboring POLEmut or MMRd—may be associated with more aggressive phenotypes, including high-grade histology, advanced FIGO stage, and lymph node metastases. These observations underscore the complexity of interpreting coexisting molecular alterations and indicate that multiple-classifier tumors may not be adequately addressed within current risk stratification systems. While our study does not provide direct prognostic evidence, the observed patterns support the need for future refinement of classification frameworks. Specifically, these tumors may require dedicated risk categories, potentially incorporating routine NGS to capture their complex molecular architecture and guide individualized therapeutic strategies.

## Figures and Tables

**Figure 1 cancers-17-02483-f001:**
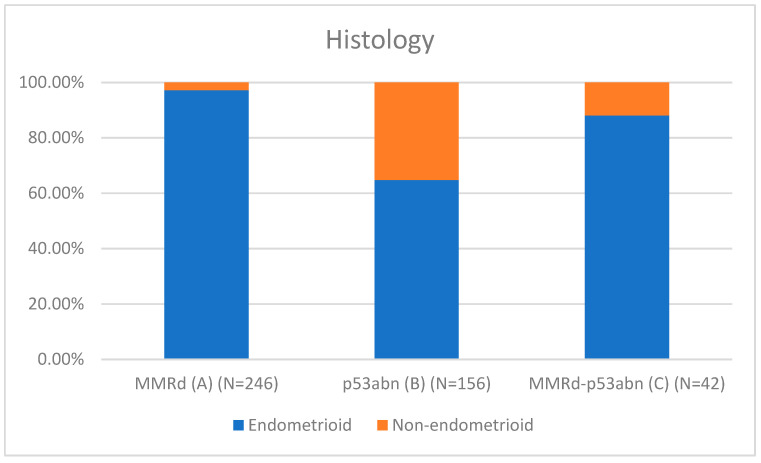
Histotype comparison.

**Figure 2 cancers-17-02483-f002:**
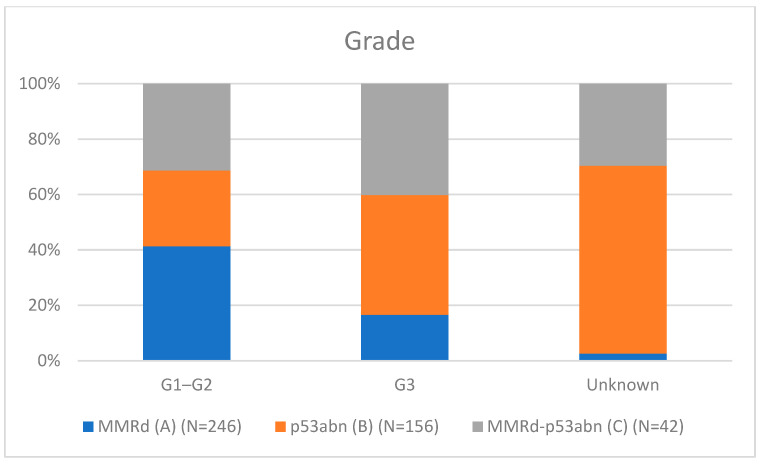
Grade comparison.

**Figure 3 cancers-17-02483-f003:**
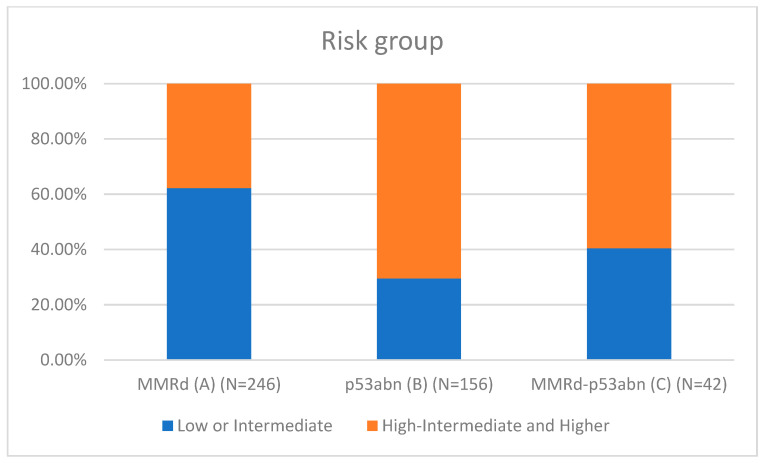
Risk group comparison.

**Table 1 cancers-17-02483-t001:** Clinicopathological characteristics of patients with endometrial cancer by molecular subtype.

Variable	Group	MMRd (A) (N = 246)	p53abn (B) (N = 156)	MMRd-p53abn (C) (N = 42)	*p**-Value (A vs. C)	OR (95% CI, A vs. C)	*p**-Value (B vs. C)	MMRd + MMRd-p53abn (N = 288)
Age at surgery (years)	Mean (SD)	66.05 (9.62)	68.19 (9.90)	67.81 (11.85)	0.291	-	0.718	66.32 (9.99%)
	<60 years	72 (29.27%)	24 (15.38%)	11 (26.19%)	0.300	-	0.098	83 (28.82%)
	60–70 years	80 (32.52%)	51 (32.69%)	9 (21.43%)				89 (30.90%)
	>70 years	93 (37.80%)	77 (49.36%)	22 (52.38%)				115 (39.93%)
	Unknown	1 (0.41%)	0 (0.00%)	0 (0.00%)				1 (0.35%)
Histology	Endometrioid	239 (97.15%)	101 (64.74%)	37 (88.10%)	0.018	3.78 (1.19–12.02)	0.002	276 (95.83%)
	Non-endometrioid	7 (2.85%)	55 (35.26%)	5 (11.90%)				12 (4.17%)
Grade	G1–G2	216 (87.80%)	91 (58.33%)	28 (66.67%)	0.002	3.26 (1.55–6.86)	0.170	244 (84.72%)
	G3	29 (11.79%)	48 (30.77%)	12 (28.57%)				41 (14.24%)
	Unknown	1 (0.41%)	17 (10.90%)	2 (4.76%)				3 (1.04%)
LVSI	Absent or focal	181 (73.58%)	102 (65.38%)	25 (59.52%)	0.054	1.88 (0.96–3.66)	0.818	206 (71.53%)
	Substantial	64 (26.02%)	51 (32.69%)	17 (40.48%)				81 (28.13%)
	Unknown	1 (0.41%)	2 (1.28%)	0 (0.00%)				1 (0.35%)
Myometrial invasion	<1/2	140 (56.91%)	81 (51.92%)	18 (42.86%)	0.091	1.81 (0.94–3.49)	0.277	158 (54.86%)
	≥1/2	105 (42.68%)	75 (48.08%)	23 (54.76%)				128 (44.44%)
	Unknown	1 (0.41%)	0 (0.00%)	1 (2.38%)				2 (0.69%)
Cervical involvement	No	177 (71.95%)	101 (64.74%)	27 (64.29%)	0.359	1.35 (0.67–2.71)	1.000	204 (70.83%)
	Yes	68 (27.64%)	55 (35.26%)	14 (33.33%)				82 (28.47%)
	Unknown	1 (0.41%)	0 (0.00%)	1 (2.38%)				2 (0.69%)
Lymph node metastases	No	208 (84.55%)	124 (79.49%)	31 (73.81%)	0.067	3.02 (1.08–8.45)	0.721	239 (82.99%)
	Yes	13 (5.28%)	16 (10.26%)	6 (14.29%)				19 (6.60%)
	Unknown	25 (10.16%)	16 (10.26%)	5 (11.90%)				30 (10.42%)
Distant metastases	No	236 (95.93%)	145 (92.95%)	41 (97.62%)	1.000	-	0.208	277 (96.18%)
	Yes	3 (1.22%)	8 (5.13%)	0 (0.00%)				3 (1.04%)
	Unknown	7 (2.85%)	3 (1.92%)	1 (2.38%)				8 (2.78%)
FIGO stage	Early (I–II)	205 (83.33%)	111 (71.15%)	31 (73.81%)	0.192	2.13 (0.96–4.72)	0.854	236 (81.94%)
	Advanced (III–IV)	34 (13.82%)	42 (26.92%)	10 (23.81%)				44 (15.28%)
	Unknown	7 (2.85%)	3 (1.92%)	1 (2.38%)				8 (2.78%)
Risk group	Low or intermediate	153 (62.20%)	46 (29.49%)	17 (40.48%)	0.001	2.81 (1.50–5.25)	0.319	170 (59.03%)
	High–intermediate and higher	93 (37.80%)	110 (70.51%)	25 (59.52%)				118 (40.97%)

Notes: Abbreviations: MMRd = mismatch repair deficient; p53abn = p53 abnormal; LVSI = lymphovascular space invasion; FIGO = International Federation of Gynecology and Obstetrics; SD = standard deviation; OR = odds ratio; CI = confidence interval. Statistical significance: * *p* < 0.05. Columns: MMRd (A): classical MMRd tumors; p53abn (B): classical p53abn tumors; MMRd-p53abn (C): multiple-classifier MMRd-p53abn tumors; MMRd + MMRd-p53abn: combined MMRd and MMRd-p53abn cases. *p*-values: calculated for comparisons between MMRd vs. MMRd-p53abn and p53abn vs. MMRd-p53abn using Fisher’s exact test for small cell counts or chi-square test for larger counts. OR (95% CI): calculated for comparisons with *p* < 0.05 (MMRd vs. MMRd-p53abn) using logistic regression for binary outcomes. Risk group assigned according to ESGO/ESTRO/ESP 2020 guidelines—molecular classification unknown. LVSI definitions: absent (0 vessels), focal (1–4 vessels), substantial (≥5 vessels).

**Table 2 cancers-17-02483-t002:** Comparison of POLEmut and multiple-classifier POLEmut tumors.

Variable	Group	POLEmut (N = 30)	POLEmut-MMRd (N = 18)	POLEmut-p53abn (N = 4)	POLEmut-MMRd-p53abn (N = 10)	*p*-Value (POLEmut vs. POLEmut-MMRd)	*p*-Value (POLEmut vs. POLEmut-p53abn)	OR (95% CI, POLEmut vs. POLEmut-p53abn)	*p*-Value (POLEmut vs. POLEmut-MMRd-p53abn)	Total (N = 62)
Grade	G3	2 (6.67%)	3 (16.67%)	3 (75.00%)	2 (20.00%)	0.198	0.005 *	42.00 (2.87–614.8)	0.247	10 (16.13%)
Lymph node metastases	Yes	1 (3.33%)	1 (5.56%)	2 (50.00%)	3 (30.00%)	1.000	0.013	29.00 (1.77–475.3)	0.192	7 (11.29%)
FIGO stage	Advanced (III–IV)	2 (6.67%)	2 (11.11%)	3 (75.00%)	4 (40.00%)	1.000	0.005 *	42.00 (2.87–614.8)	0.033 *	11 (17.74%)

Notes: Abbreviations: POLEmut = POLE ultramutated; MMRd = mismatch repair deficient; p53abn = p53 abnormal; FIGO = International Federation of Gynecology and Obstetrics; OR = odds ratio; CI = confidence interval. Statistical significance: * *p* < 0.05. Data presentation: values are presented as n (%) unless otherwise specified. Statistical tests: Fisher’s exact test for categorical variables (small cell counts). Unknown categories excluded from statistical tests. *p*-values: calculated for comparisons between POLEmut and each multiple-classifier subgroup. OR and 95% CI: calculated for POLEmut vs. POLEmut-p53abn. Risk group assigned according to ESGO/ESTRO/ESP 2021 guidelines, incorporating molecular classification per ProMisE.

## Data Availability

The original contributions presented in this study are included in this article. Further inquiries can be directed to the corresponding author(s).

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
