# Peer review of "Clinicopathological Features and Risk Stratification of Multiple-Classifier Endometrial Cancers: A Multicenter Study from Poland"

_cancers, 2025, doi:10.3390/cancers17152483_

Round 1
Reviewer 1 Report
Comments and Suggestions for Authors
In the present paper, the authors report the “Clinicopathological Features and Risk Stratification of Multi-2 ple-Classifier Endometrial Cancers: A Multicenter Study from 3 Poland”. I would recommend the acceptance of this manuscript after minor revisions. Here are the following comments;
Comments
- How can we determine the aggressiveness or prognosis of POLEmut-p53abn and MMRd-p53abn ECs without knowing overall survival (OS) or progression-free survival (PFS)?
- Given small sample sizes for crucial subgroups like POLEmut-p53abn (N=4) and triple classifiers (N=10), how do you justify the statistical significance of your findings, especially with such large confidence intervals?
- What was the reason for not employing multivariate regression to address confounding factors like age, stage, and comorbidities in the analysis of the correlation between molecular subtypes and clinicopathologic features?
- Why did you not categorize MMRd cancers depending on their underlying cause, such as Lynch syndrome vs. MLH1 promoter methylation, which could have a substantial impact on biological behavior and treatment outcome?
- What rationale supports the external validity of your findings, given that all patients came from Polish oncology centers and potential genetic or diagnostic differences may limit broader applicability?
- In the absence of a central pathology review, what measures can the four participating centers implement to ensure consistency and reliability in their evaluations of histological subtype, tumor grade, and LVSI?
Author Response
Dear Reviewer 1,
We sincerely thank you for your thorough review and positive evaluation of our manuscript, ‘Clinicopathological Features and Risk Stratification of Multiple-Classifier Endometrial Cancers: A Multicenter Study from Poland’ (Manuscript ID: cancers-3754248). Your recommendation for acceptance after minor revisions and your insightful comments have greatly helped us improve the clarity and scientific rigor of our work. Below, we address each of your comments point-by-point and describe the revisions made to the manuscript.
- How can we determine aggressiveness or prognosis for POLEmut-p53abn and MMRd-p53abn tumors without overall survival (OS) or progression-free survival (PFS) data?
Response:
We fully acknowledge that the absence of OS and PFS data is a significant limitation of our study, as stated in the "Limitations" section. The assessment of tumor aggressiveness in POLEmut–p53abn and MMRd–p53abn subgroups was based on well-established clinicopathological risk factors, including high tumor grade (G3: 28.6% vs. 11.8% in MMRd;p= 0.002) and advanced stage (FIGO III–IV: 75% in POLEmut–p53abn vs. 6.7% in POLEmut; p = 0.005). These features are widely recognized as indicators of unfavorable prognosis in endometrial cancer, as supported by prior studies (e.g., Kato [18], Michalova [21]). Nevertheless, we emphasize that our findings should be interpreted with caution, and we strongly support the need for further prospective studies including survival data to validate these observations.
As noted in section 3.3 of the manuscript, survival data were not available due to the relatively short follow-up period (2022–2025).
- Given small sample sizes for crucial subgroups like POLEmut-p53abn (N=4) and triple classifiers (N=10), how do you justify the statistical significance of your findings, especially with such large confidence intervals?
Response:
The small size of subgroups (especially POLEmut-p53abn and POLEmut-MMRd-p53abn) limits statistical power and results in wide confidence intervals. Throughout the manuscript, we repeatedly emphasize cautious interpretation and consider our findings as signals that require validation in larger cohorts. Statistical significance here is exploratory and should not be interpreted as definitive evidence.
- What was the reason for not employing multivariate regression to address confounding factors like age, stage, and comorbidities in the analysis of the correlation between molecular subtypes and clinicopathologic features?
Response:
As noted, multivariate regression was not performed in this study due to the small size of several molecular subgroups and the absence of complete data on comorbidities. Moreover, age did not differ significantly between the analyzed groups (e.g., MMRd vs. MMRd–p53abn: p = 0.291). We opted for direct comparisons using Fisher's exact and chi-square tests to preserve interpretability in this heterogeneous dataset. We have now clarified this limitation in the revised manuscript, under the “Limitations” section, and plan to incorporate multivariate models in future studies with larger and more homogeneous cohorts.
- Why did you not categorize MMRd cancers depending on their underlying cause, such as Lynch syndrome vs. MLH1 promoter methylation, which could have a substantial impact on biological behavior and treatment outcome?
Response:
Detailed subclassification of MMRd tumors was not feasible due to retrospective data limitations and the lack of complete methylation testing across centers, as noted in the “Limitations.” We focused on MMR protein loss (IHC), consistent with ProMisE criteria. In future studies, we plan to analyze MMRd subtypes to assess their biological behavior and therapeutic implications (e.g., immunotherapy).
- What rationale supports the external validity of your findings, given that all patients came from Polish oncology centers and potential genetic or diagnostic differences may limit broader applicability?
Response:
The Polish population may limit the generalizability of the results. However, our aim was to present the first multicenter analysis of multiple-classifier ECs from Central-Eastern Europe, a region underrepresented in the literature. Molecular classifiers (MMRd, p53abn, POLEmut) are based on globally accepted criteria (e.g., ProMisE), which enhances comparability. Results should still be interpreted cautiously across different populations, and further international studies are needed.
- In the absence of a central pathology review, what measures can the four participating centers implement to ensure consistency and reliability in their evaluations of histological subtype, tumor grade, and LVSI?
Response:
The absence of central pathology review (noted in the “Limitations”) could be a limitation. However, each center followed uniform WHO 2020 guidelines, and pathologists maintained regular communication and used harmonized interpretative criteria, which helped minimize potential variability in assessments.
Reviewer 2 Report
Comments and Suggestions for Authors
This is an interesting multicenter study across four Polish oncology centers to evaluate the prevalence and clinicopathological features of multiple-classifier endometrial cancers. The paper reports a real-world experience based on a significant number of cases. In would like to suggest some comments on this manuscript-
a) line 111 " In selected cases with discordant or inconclusive findings ...". Which were the parameters to consider the assays discordant or inconclusive. It would be better to specify-
b) line 112 " (NGS) was conducted on the IonTorrent platform using AmpliSeq panels..." Please specify which panels: commercially available tumor specific or custom panels
c) The Authors cite the paper of Leon Castillo et al. and suggest a comparison between the prognostic parameters based on clinical follow-up proposed in J Pathol 2020 and pathological features ( non endometrioid histology and grading) and risk status per ESGO/ESTRO/ESP. These are not comparable parameters. The molecular classification has been proposed to overcome the limits of the traditional parameters of risk stratification
d) line 239 and following. It has been reported that the presence of multiclassifier POLE-p53 may exhibit an intermediate behaviour between POLE and p53 abnormal tumors. However it would be premature to draw conclusions from a limited number of cases without a significant follow-up
e) line 245 and following. The Authors should discuss that some discrepancies in defining p53 status can be due to different methods of detection. There is a higher incidence of alterations identified with sequencing than with immunohistochemistry. Another point to discuss is the presence of subclonal immunostaining. The Authors report it in "Methods" but not in "Results" and in "Discussion". A comment should be done since many difficulties in evaluating the p53 status arise in these cases. Explanations for discordant results for p53 immunohistochemistry and NGS include exon splice sites or large scale deletions or insertions (indels), often missed with a targeted NGS assay.
f) The analysis of POLE exonuclease domain mutations with Sanger sequencer is a low cost opportunity to evaluate the status of a single gene. The Authors might comment this choice in the NGS era. Another technique that could offer more sensibility mantainig a low cost is RT-PCR. There is an interesting report from Poland on Sanger sequencing for POLE evaluation : Laczmanska I, Michalowska D, Jedryka M, Blomka D, Semeniuk M, Czykalko E, Abrahamowska M, Mlynarczykowska P, Chrusciel A, Pawlak I, Maciejczyk A. Fast and reliable Sanger POLE sequencing protocol in FFPE tissues of endometrial cancer. Pathol Res Pract. 2023 Feb;242:154315. doi: 10.1016/j.prp.2023.154315. Epub 2023 Jan 18. PMID: 36738508.
g) Which was the clinical impact of this diagnostic effort on the clinical approach to endometrial cancer in the centers involved in the study?
Author Response
Dear Reviewer 2,
We are deeply grateful for your thoughtful review and kind words regarding our manuscript, ‘Clinicopathological Features and Risk Stratification of Multiple-Classifier Endometrial Cancers: A Multicenter Study from Poland’ (Manuscript ID: cancers-3754248). Your recognition of the study’s value as an interesting multicenter analysis and your constructive suggestions have been instrumental in enhancing the manuscript’s quality. We have carefully addressed each of your comments below, outlining the revisions made to ensure clarity and alignment with the study’s objectives.
a) Line 111: In selected cases with discordant or inconclusive findings ...". Which were the parameters to consider the assays discordant or inconclusive. It would be better to specify-
Response:
We have updated the “Methods” section (line 111): “Discordant or inconclusive findings were defined as ambiguous MMR or p53 IHC staining (e.g., partial loss or borderline subclonal p53), discrepancies with clinicopathological features, or suspected multiple-classifier profiles requiring validation.”
b) line 112 " (NGS) was conducted on the IonTorrent platform using AmpliSeq panels..." Please specify which panels: commercially available tumor specific or custom panels
Response:
We used a custom-designed AmpliSeq panel (Thermo Fisher Scientific) targeting POLE, TP53, MLH1, MSH2, MSH6, and PMS2. We have updated the “Methods” section (line 112): “NGS was performed on the IonTorrent platform using a custom-designed AmpliSeq panel targeting POLE, TP53, MLH1, MSH2, MSH6, and PMS2 genes.”
c)The Authors cite the paper of Leon Castillo et al. and suggest a comparison between the prognostic parameters based on clinical follow-up proposed in J Pathol 2020 and pathological features ( non endometrioid histology and grading) and risk status per ESGO/ESTRO/ESP. These are not comparable parameters. The molecular classification has been proposed to overcome the limits of the traditional parameters of risk stratification
Response:
We agree with the reviewer that molecular classification aims to address the limitations of traditional pathological parameters, and direct comparisons of prognostic outcomes (e.g., recurrence-free survival) with clinicopathological features are not appropriate. In our study, we did not intend to compare prognostic outcomes but rather to highlight that MMRd-p53abn endometrial cancers (ECs) exhibit a distinct morphological profile compared to MMRd-only tumors. To address this concern, we have revised the ‘Discussion’ section to clarify that our findings focus on clinicopathological differences, such as non-endometrioid histology (11.9% vs. 2.85%, p=0.018), grade 3 tumors (28.6% vs. 11.8%, p=0.002), and high-intermediate/high-risk status (59.5% vs. 37.8%, p=0.001), which suggest a more aggressive profile for MMRd-p53abn tumors. These observations are presented as hypothesis-generating, and we emphasize the need for future studies with survival data to evaluate their prognostic implications.
d)line 239 and following. It has been reported that the presence of multiclassifier POLE-p53 may exhibit an intermediate behaviour between POLE and p53 abnormal tumors. However it would be premature to draw conclusions from a limited number of cases without a significant follow-up
Response:
We agree, and we have already highlighted this caution in the discussion and the “Limitations” section. Therefore, no additional changes were made to the text.
e)line 245 and following. The Authors should discuss that some discrepancies in defining p53 status can be due to different methods of detection. There is a higher incidence of alterations identified with sequencing than with immunohistochemistry. Another point to discuss is the presence of subclonal immunostaining. The Authors report it in "Methods" but not in "Results" and in "Discussion". A comment should be done since many difficulties in evaluating the p53 status arise in these cases. Explanations for discordant results for p53 immunohistochemistry and NGS include exon splice sites or large scale deletions or insertions (indels), often missed with a targeted NGS assay.
Response:
We agree that discrepancies in p53 status may arise from using different diagnostic methods. In our study, p53 status was primarily assessed via immunohistochemistry (IHC), with subclonal staining (defined as focal, heterogeneous expression) observed in 11.9% of MMRd-p53abn cases.
We added to the discussion: “Discrepancies in p53 status between IHC and NGS may arise from splice-site mutations, large indels, or subclonal staining (observed in 11.9% of MMRd-p53abn cases), often missed by targeted NGS. These challenges highlight the need for integrated IHC-NGS approaches to improve classification accuracy.”
f) The analysis of POLE exonuclease domain mutations with Sanger sequencer is a low cost opportunity to evaluate the status of a single gene. The Authors might comment this choice in the NGS era. Another technique that could offer more sensibility mantainig a low cost is RT-PCR. There is an interesting report from Poland on Sanger sequencing for POLE evaluation : Laczmanska I, Michalowska D, Jedryka M, Blomka D, Semeniuk M, Czykalko E, Abrahamowska M, Mlynarczykowska P, Chrusciel A, Pawlak I, Maciejczyk A. Fast and reliable Sanger POLE sequencing protocol in FFPE tissues of endometrial cancer. Pathol Res Pract. 2023 Feb;242:154315. doi: 10.1016/j.prp.2023.154315. Epub 2023 Jan 18. PMID: 36738508.
Response:
Sanger sequencing was applied as a standard, widely available, and well-established method for analyzing POLE exonuclease domain mutations, consistent with long-standing practice in most Polish centers. In our study, NGS was also used when further verification was needed (e.g., inconclusive results, discordance with pathological features, suspected additional mutations). We agree that Sanger sequencing remains cost-effective and valuable for single-gene analysis, as confirmed by the report from Laczmanska et al., which we cited. RT-PCR, while potentially more sensitive in certain contexts, was not routinely used in our centers for POLE mutation detection.
g) Which was the clinical impact of this diagnostic effort on the clinical approach to endometrial cancer in the centers involved in the study?
Response:
Our study was primarily exploratory and did not directly impact treatment protocols during the study period. However, the results were discussed in multidisciplinary tumor boards, which raised awareness among clinical teams that molecular classification is not always straightforward and may require a more individualized approach. This emphasized the need for cautious implementation of new classification systems, especially in the absence of large prospective studies validating their prognostic and predictive value. We believe that our work can contribute to further evaluation of molecular risk stratification and support the development of more personalized treatment strategies in the future.
Round 2
Reviewer 2 Report
Comments and Suggestions for Authors
The Authors discussed the proposed suggestions and modified the text when necessary.
The paper described an important, local effort to apply a molecular classification with simple and widely available techniques.
Author Response
Dear Reviewer 2,
We are deeply grateful for your thoughtful review and kind words regarding our manuscript, ‘Clinicopathological Features and Risk Stratification of Multiple-Classifier Endometrial Cancers: A Multicenter Study from Poland’ (Manuscript ID: cancers-3754248). Your recognition of the study’s value as an interesting multicenter analysis and your constructive suggestions have been instrumental in enhancing the manuscript’s quality.